# A Novel Button-Type Micro Direct Methanol Fuel Cell with Graphene Diffusion Layer

**DOI:** 10.3390/mi10100658

**Published:** 2019-09-29

**Authors:** Yingli Zhu, Lei Gao, Jianyu Li

**Affiliations:** Tianjin Key Laboratory of Integrated Design and On-line Monitoring for Light Industry & Food Machinery and Equipment, School of Mechanical Engineering, Tianjin University of Science & Technology, Tianjin 300384, China; 18853561233@163.com (L.G.); lijianyu@tust.edu.cn (J.L.)

**Keywords:** micro direct methanol fuel cells (μDMFCs), button-type, graphene, gas diffusion layer (GDL)

## Abstract

In order to solve the problem that bolts in traditional packaged direct methanol fuel cells (DMFCs) take up a large area and reduce the specific energy (energy per unit weight) and power density (power per unit area), a new button-type micro direct methanol fuel cell (B-μDMFC) is designed, assembled, and packaged. The cell with four different structures was tested before and after packaging. The results indicate that the button cell with three-dimensional graphene and springs has the best performance. The equivalent circuit and methanol diffusion model was applied to explain the experimental results. The peak volumetric specific power density of the cell is 11.85 mW cm^−3^. This is much higher than traditional packaged DMFC, because the novel B-μDMFC eliminates bolts in the structure and improves the effective area ratio of the cell.

## 1. Introduction

Close attention [1,2,3] has been paid to the micro direct methanol fuel cell (μDMFC), because it is the potential mobile energy for portable devices [4,5,6,7]. The packaging technology of μDMFC affects the internal resistance of the cell and the specific energy of the cell, thereby affecting the overall output performance of the cell. It is necessary to ensure the components are in good contact and have high porosity of the diffusion layer. Most packaging methods are mechanical bonding, hot pressing, and high polymer bonding. Takahiro Shimizu [8] reported a self-breathing direct methanol fuel cell packaged by bolting plexiglass. Kyong-Bok Min [9], J. Yeom [10], and Guo Zhen [11] encapsulated fuel cells by thermo-compression bonding, but the output performance of the cell was relatively poor. It is difficult to form a strong bonding layer in the interface and reduce the contact resistance with the method of thermo-compression bonding. Thus, it limits the output performance of the cell. Therefore, the packaging of traditional block-type fuel cells usually requires a lot of bolts for moderate compression pressure to obtain lower ohmic resistance and mass transfer resistance. The existence of bolts, however, may consume a large volume of the cell and reduce the power density per unit volume, especially for micro cells.

In order to solve this problem, a new button-type micro direct methanol fuel cell (B-μDMFC) is designed, assembled, and packaged. The design of the B-μDMFC is based on the layout of the traditional block-type direct methanol fuel cell and that of the button-type zinc-air batteries [12]. In addition, this paper especially compares the performances of B-μDMFCs with different material structures and analyzes the mechanism using the equivalent circuit and diffusion model.

## 2. Experimental

The proton exchange membrane (PEM) of the B-μDMFCs was DuPont’s Nafion 117 membrane, with a thickness of 183 μm. The catalyst layer was applied on both sides of the Nafion 117 membrane. The anode catalyst loading was 4 mg·cm^−2^ carbon-supported 1:1 Pt/Ru, and the cathode side was 2 mg·cm^−2^ carbon-supported Pt. The PEM diameter was 1.94 cm, as shown in Figure 1. The sealing silica gel gaskets with a thickness of 0.2 mm are regarded as insulators. Two materials were applied as the gas diffusion layer (GDL) of the cell in this paper. One was three-dimensional graphene (3DG) based on foam nickel(FN) [13] (Shenzhen Six Carbon Technology Co., Ltd., Shenzhen, China) and the other was TGP-090 carbon paper (CP) (Toray Co., Ltd., Tokyo, Japan). The diameter of the titanium-plated stainless steel mesh was 15 mm. Two types of supporting conductive structure (FN and stainless steel springs) were used as current collectors (CCs) in the B-μDMFC.

The components and structure of the button-type micro direct methanol fuel cell (B-μDMFC) are shown in Figure 1 and Figure 2, respectively. Similar to the conventional DMFC, the key component of the cell is a membrane electrode assembly (MEA), which comprised a proton exchange membrane, cathode and anode catalytic layer, and cathode and anode gas diffusion layer. The diameter of the active area of the MEA was 1.5 cm, and thus the area was 1.77 cm^2^. Carbon paper and three-dimensional graphene were adopted as the cathode and anode GDL, respectively. In order to reduce the internal contact resistance of the cell and support the GDL, a titanium-plated stainless steel wire mesh was added between the gas diffusion layer and the end plate on both sides of the anode and cathode. The lid and cup of standard lithium battery were used as the anode and the cathode end plate of the B-μDMFC.

Finally, the B-μDMFCs were assembled as four structures: three-dimensional graphene–springs (3DG + S), carbon paper–springs (CP + S), 3D graphene–foam nickel (3DG + FN), and carbon paper–foam nickel (CP + FN). It was reported that the optimal packaging pressure of the DMFC was 0.5 MPa [14,15]. Considering the plastic deformation of the components, the pressure of our experiments was 1 MPa. Figure 3 shows the packaged 3DG + S B-μDMFC. Table 1 shows the physical properties of four B-μDMFCs.

A methanol reservoir with the depth of 2.5 mm was supported by springs in the lid of the cell with springs. The springs for providing and maintaining the compression pressure on the cell were made of 304 stainless steel with good elasticity and low internal resistance. Foamed nickel was also used to replace the springs. A gas-liquid separation membrane was set in the hole in the lid to facilitate of carbon dioxide removal and methanol injection. 

## 3. Results and Discussion

### 3.1. Physical Characterization of GDL

Figure 4a is a low-magnification scanning electron microscope (SEM) image of 3DG to show the 3DG interconnected network structure in order to provide a large specific surface area and high porosity. This structure provides a multi-channel pathway for methanol diffusion and CO_2_ removal. 3DG is also an excellent current collector owing to its high conductivity. Figure 4b is a high-magnification SEM image of 3DG to suggest that graphene was uniformly coated on a FN substrate. The graphene film was continuously grown based on chemical vapor deposition(CVD) [13]. In addition, because of the difference in thermal expansion between the FN and the grown graphene film, irregular wrinkles of the graphene surface can be seen in Figure 4b. Figure 4c,d are SEM images of carbon paper and micro-porous layers, respectively. As can be seen from the figure, the Teflon is uniformly covered on carbon fibers, so the water generated in the cell can be eliminated timely. The dense structure of the carbon fiber cross-section is beneficial for the improvement of electrical conductivity, but may cause a lower porosity for being not conducive to the transmission of liquid and gas.

The 3DG has a big deformation after B-μDMFCs’ assembly. The thickness of 3DG adjusted to 165 μm from 495 μm under the pressure of 1 MPa and, consequently, the porosity changed from the initial 90% to 70%, as shown in Figure 4 and Figure 5.

### 3.2. Cell Performance

The B-μDMFCs were packaged in a packaging machine for lithium-ion button battery. Before assembly, the cells were tested in a press machine under a constant pressure of 1 MPa. Both of the tests of the B-μDMFC before and after packaging were carried out at room temperature and in passive mode. The methanol concentration used in this experiment was 1 mol L^−1^. As shown in Figure 6a, the peak powers of all four cells exceed 10 mW. The peak power of the CP + S button cell is 12 mW, which is the lowest of the four cells. The 3DG + FN cell has the highest peak power of 16.5 mW, which means that the cell discharge current can reach 82 mA when the voltage is rated at 0.2 V. Figure 6b shows the Nyquist curve of various structural button fuel cells before packaging. The 3DG + FN button fuel cells have a minimum ohm resistance of 0.4 Ω and the CP + S button fuel cells have a maximum ohm resistance of 0.8 Ω.

After packaging, the output performances of all button cells decreased because of the excessive pressure and elastic recovery of the shell of the cell after removing the pressure from the packaging machine. This phenomenon was also illustrated in our previous work because of the plastic deformation of the GDL [14]. The peak power of the 3DG + S structural cell changed from 14.5 mW to 14.2 mW, which was the smallest change of the output power. The ohmic internal resistance changed from 0.46 Ω to 0.48 Ω and the maximum current density of the 3DG + S structural B-μDMFC reached 34 mA cm^−2^. The performances of the cells with the other three structures became worse. The performance of the button cell dropped sharply, especially when the foamed nickel (FN) was used as the support structure. The peak power of the 3DG + FN structural fuel cell was 10.75 mW, and the ohmic internal resistance reached 0.89 Ω. The performance of the CP + FN cell was the worst. The peak power was only 7.5 mW and the ohm internal resistance was 1.16 Ω. This is because the FN cannot store elastic restoring force and has poor contact with the neighbor components, which consequently leads to a large internal resistance of the cell. It can be seen from Figure 5 that the cell performance of 3DG as a GDL for B-μDMFC is much better than CP, whether the support current collecting structure is springs or FN. Table 2 shows that the B-μDMFC has higher power density than some other type DMFCs, especially for power density per unit volume.

### 3.3. Equivalent Circuit Fitting

Considering the complex reaction processes of substances (methanol, oxygen, and water), mass transport and diffusion, and electron transfer during the operation of B-μDMFC, the μDMFC full cell equivalent circuit (Randles circuit) model [18] was applied to understand the measured cell alternating current(AC) impedance. The measured Bode and Nyquist curves by electrochemical impedance spectroscopy(EIS) technology were fitted using the equivalent circuit model. As shown in Figure 7, *R*_m_ is the ohmic resistance of the cell, *R*_ct_ is the charge transferresistance, *R*_mt_ is the mass transfer resistance, *L* is the pseudo-inductanceinduced by the outside circuits and test equipments, *Q*_1_ and *Q*_2_ express the charge and discharge process of the doublelayer capacity of the anode and cathode, and resistance of *R*_CO_ and lowfrequency impedance of *L*_CO_ express the relaxation process of the CO product in the anode electrode.

As shown in Table 3, the *R*_m_ of the 3DG + S structural fuel cell is the smallest and the mass transfer resistance is the second-lowest. This is because that the porosity of the 3DG (70%) is higher than CP (53%) after assembly, and because FN has plastic deformation than springs. The difference in contact resistance (a part of *R*_m_) after assembly may also be the cause of the difference in *R*_m_ between spring and FN. Therefore, the methanol can distribute more evenly in the catalytic layer and CO_2_ can release from the cell anode in time in the B-μDMFC of 3DG + S. The *R*_m_ of the spring-structured cell is smaller than the foamed nickel-structured cell. This is because the spring has good elasticity and the components can be in close contact after being pressed, and the FN has poor elasticity and cannot be recovered after being compressed.

## 4. Anode Methanol Diffusion Model

### 4.1. Deformation Analysis

The support current collector (FN) and the gas diffusion layer (CP and 3DG) in the cell are porous materials and will be greatly deformed under compression pressure. Large deformation often causes lower porosity of porous materials, which has a great influence on the cell performance. The support current collector FN used herein has a thickness of 4 mm and the thickness of 3DG is 0.5 mm. The geometric model (shown in Figure 8) consists of the cell lid, anode support current collector, and anode gas diffusion layer (3DG). A uniform pressure of 1 MPa is applied to the cell lid, and the bottom of the GDL is fixed.

In order to verify the accuracy of the simulation, compression experiments of FN and 3DG were also performed. After applying a pressure of 1MPa to the FN, the total thickness deformation was 1.52 mm, which was close to the simulated results of 1.39 mm. For 3DG, the thickness of 3DG became 0.16 mm under the pressure of 1 MPa. The void ratio of FN and 3DG after compression deformation can be estimated by the following formula.
(1)$$\varepsilon =1-\frac{d_{0}}{d}(1-\varepsilon_{0}),$$
where *ε* is the current porosity of the material, *ε*_0_ is the initial porosity before compression (3DG is about 90%, FN is about 75%), *d* is the current thickness, and *d*_0_ is the initial thickness (3DG is 0.5 mm, FN is 4 mm). After being applied by 1MPa pressure, the porosity of the FN changed to 59% and the porosity of the 3DG changed from the initial 90% to 70%. If non-uniform compression of the diffusion layer is taken into account, the actual local porosity of the diffusion layer (3DG) under the current collector rib is even lower.

### 4.2. Methanol Concentration Diffusion Model

In order to better understand the flow distribution of methanol in the anode of the cell, a three-dimensional steady-state mathematical model was developed. In the model, the main focus is on the distribution of methanol concentration in the anode support current collector (FN) and gas diffusion layer (3DG). To simplify the model, several assumptions for the purpose of simplifications are introduced: (1) the B-μDMFC is in steady-state operation; (2) the formation of CO_2_ on the anode side of the B-μDMFC does not affect the methanol diffusion process in the diffusion layer; (3) the support current collector (FN) and gas diffusion layer (3DG) are continuous porous media; and (4) the porosity is uniform in every component.

As there was no electrochemical reaction in the anode diffusion layer, the convection-diffusion equation was used to characterize the methanol concentration distribution in the support current collector (FN) and GDL (3DG).
(2)$$\nabla \cdot (-D_{i}\nabla C_{i}+uC_{i})=R,$$
where *C* is the methanol concentration, *R* is the rate of methanol consumption at the interface between the diffusion layer and the catalytic layer, and *D* is the effective diffusion coefficient of methanol in the diffusion layer. Because of the large mass transfer resistance of the methanol in the anode diffusion layer, the effective coefficient of the aqueous methanol solution was corrected by the Bruggeman model as the following formula:(3)$$D=D_{M}\cdot \varepsilon^{1.5},$$
where *ε* is the diffusion layer porosity and *D*_M_ is the diffusion coefficient of methanol in water. The calculation model is shown in Figure 8. The boundary conditions were set as follows: *C* = *C*_0_ in methanol inlet, where *C*_0_ is the fuel concentration in the methanol chamber; *R* = −*M*_MeOH_ on the contact surface of GDL with the catalytic layer, where *M*_MeOH_ is the methanol consumption flux as the following formula:(4)$$M_{\mathrm{MeOH}}=\frac{I}{6F}+M_{\mathrm{MeOH},\mathrm{cross}}.$$

The first term at the right end of the formula is the methanol flux consumed by the current *I*, the second term is the methanol permeate flux, and *F* is the Faraday constant.

The small contact area between the spring and the GDL can be ignored. A quarter of the geometric model was chosen to calculate (shown in Figure 9) and the parameters used in the computation are shown in Table 4. As shown in Figure 10a,b, even though the porosity of the GDL (3DG) of the 3DG + S structural cell was reduced after applying clamp force, the methanol diffusion was relatively uniform in the GDL. When the internal fuel concentration was 1 M, the lowest methanol concentration at the interface between the 3DG and the catalytic layer was 0.92 M. By comparison, the lowest methanol concentration of the 3DG + FN structural cell was about 0.8 M, as shown in Figure 9c,d. This is because the porosity of FN changed from 0.75 to 0.59 after being pressed. Moreover, the methanol concentration under the FN rib was lower than that in the channel. Owing to the large contact area between FN and 3DG and the lower porosity of FN, the consumption of methanol between the 3DG and the catalytic layer made the methanol distribution uneven. This leads to a large mass transfer resistance *R*_mt_ (11.6 Ω, in Table 2) and poor cell performance.

## 5. Conclusions

A new button-type micro direct methanol fuel cell is designed and manufactured and the performance of the cell is compared with the traditional DMFC. It is found that the performance of the B-μDMFC with 3DG as a GDL is better than that with CP. The performance of the cell with spring is better than that with foamed nickel. Equivalent circuit was applied to fit the mass resistance and ohmic resistance of the cell. The methanol concentration diffusion was also simulated to explain the results. The results indicate that the B-μDMFC cell with three dimension and springs had the best performance, because it had relative low ohmic resistance and mass transfer resistance. The volumetric specific power density of the packaged B-μDMFC with 3DG + S was 11.85 mW cm^−3^ (34 mA cm^−2^), which is four times higher than that of the air-breathing μDMFC [15], and is two times higher than that of the passive DMFC [16].

## Figures and Tables

**Figure 1 micromachines-10-00658-f001:**
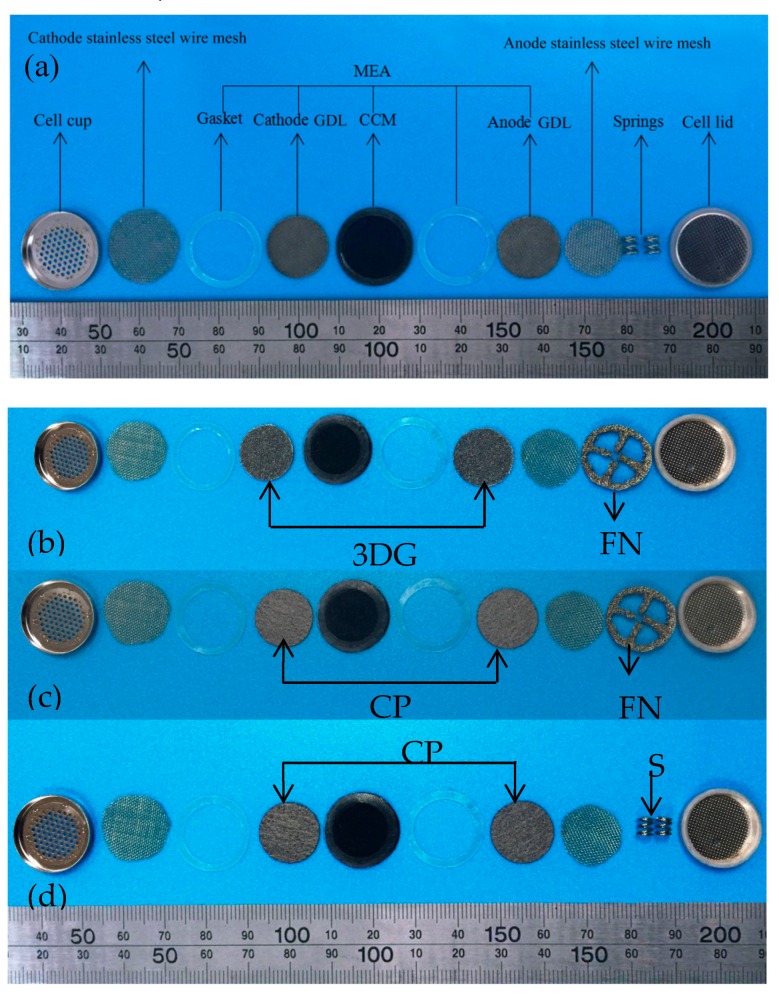
Photograph of the button-type micro direct methanol fuel cell(B-μDMFC) components: (**a**) the three-dimensional graphene–springs (3DG + S) B-μDMFC; (**b**) the 3D graphene–foam nickel (3DG + FN) B-μDMFC; (**c**) the carbon paper–foam nickel(CP + FN) B-μDMFC; and (**d**) the carbon paper–springs(CP + S) B-μDMFC. MEA, membrane electrode assembly; GDL, gas diffusion layer.

**Figure 2 micromachines-10-00658-f002:**
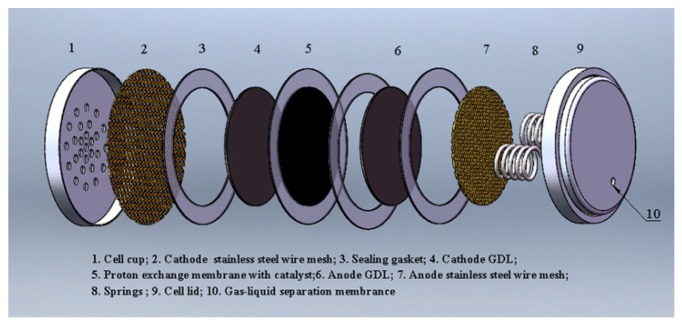
The layout of the 3DG + S B-μDMFC.

**Figure 3 micromachines-10-00658-f003:**
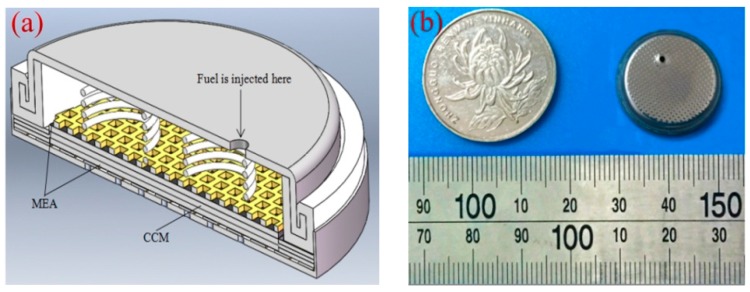
(**a**) Cross-sectional view of the 3DG + S B-μDMFC; (**b**) the packaged B-μDMFC.

**Figure 4 micromachines-10-00658-f004:**
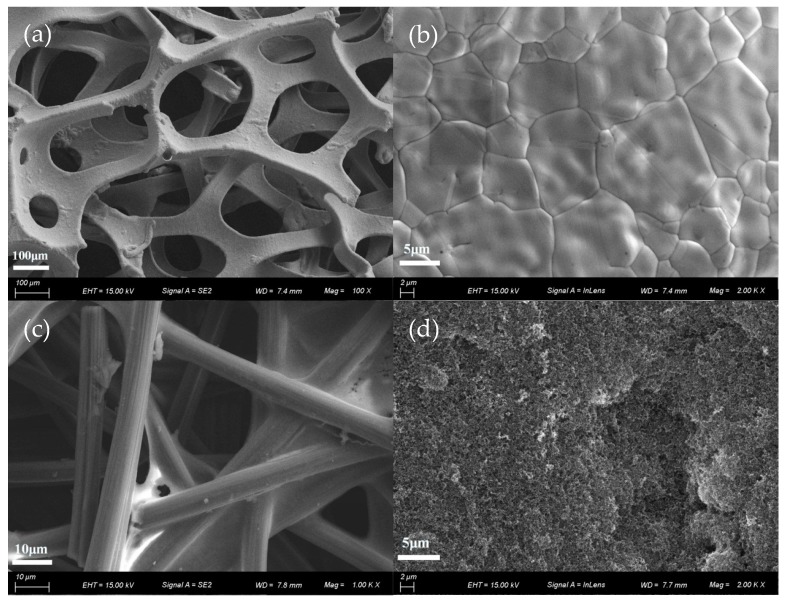
(**a**,**b**) Scanning electron microscope (SEM) image of 3DG; (**c**) anode of CP; (**d**) microporous layer of CP.

**Figure 5 micromachines-10-00658-f005:**
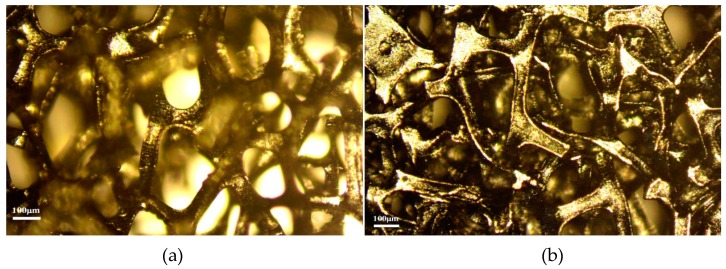
The view of 3DG: (**a**) top view before assembly; (**b**) top view after assembly; (**c**) cross-sectional view before assembly; (**d**) cross-sectional view after assembly.

**Figure 6 micromachines-10-00658-f006:**
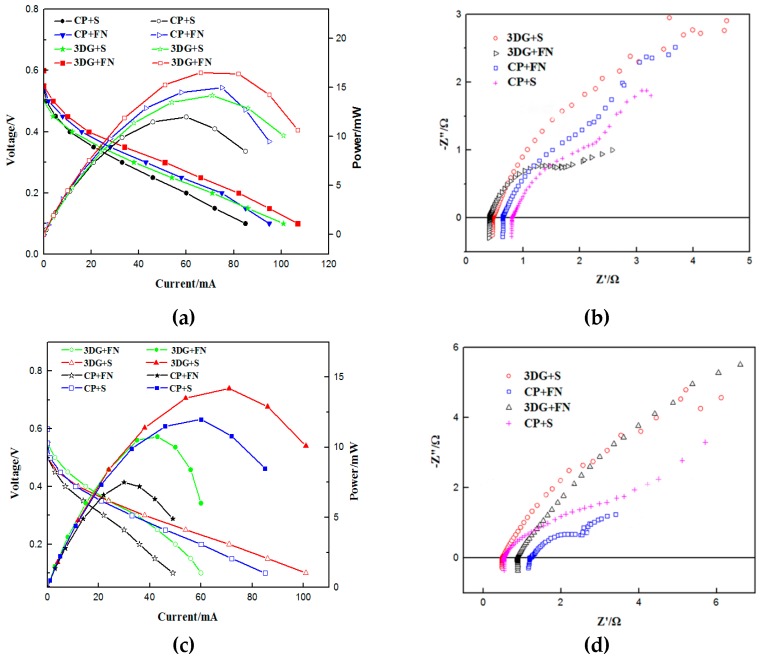
The polarization curve of various B-μDMFCs before packaging (**a**) and after packaging (**c**), and the Nyquist curve of various B-μDMFCs before packaging (**b**) and after packaging (**d**).

**Figure 7 micromachines-10-00658-f007:**
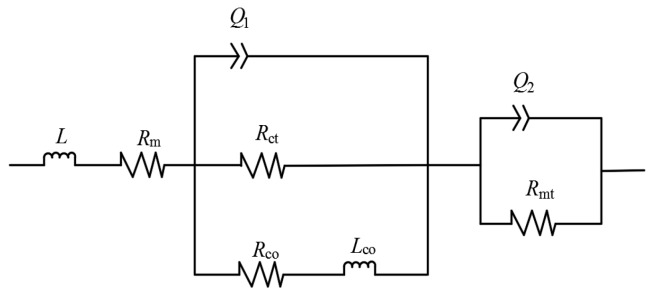
Equivalent circuit (Randles circuit).

**Figure 8 micromachines-10-00658-f008:**
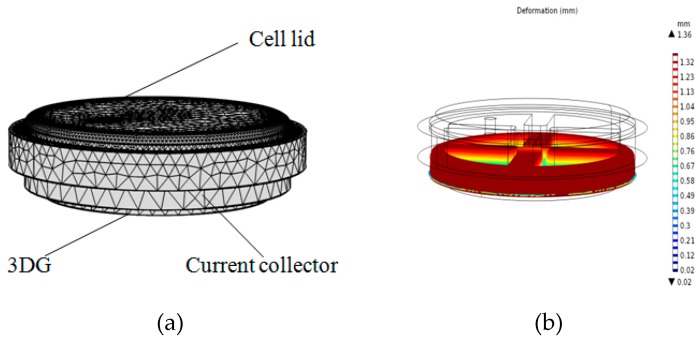
(**a**) The calculation model; (**b**) the deformation of support current collector (FN).

**Figure 9 micromachines-10-00658-f009:**
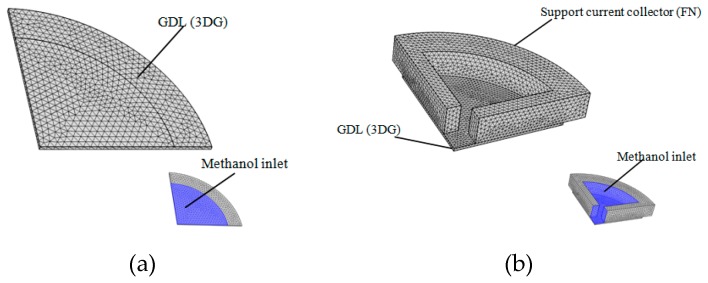
(**a**) The methanol concentration diffusion model of the 3DG + S cell; (**b**) three-dimensional methanol concentration diffusion model of the FN + 3DG cell.

**Figure 10 micromachines-10-00658-f010:**
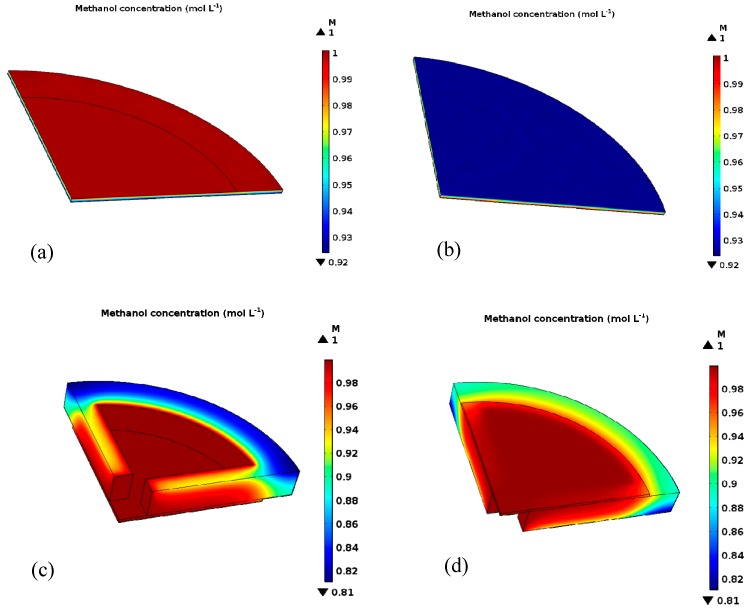
(**a**) Front view of methanol concentration distributions in the gas diffusion layer (GDL) of the 3DG + S structural cell; (**b**) back view of methanol concentration distributions in GDL of the 3DG + S structural cell; (**c**) front view of methanol concentration distributions in the 3DG + FN structural cell; (**d**) back view of methanol concentration distributions in the 3DG + FN structural cell.

**Table 1 micromachines-10-00658-t001:** Physical properties of the fourbutton-type micro direct methanol fuel cells(B-μDMFCs). 3DG, three-dimensional graphene; S, springs; FN, foam nickel; CP, carbon paper; GDL, gas diffusion layer; CC, current collector.

Cell Type	GDL	CC	Thickness or Specificationof CC (mm)	Porosity of GDL	Thickness of GDL (mm)
3DG + S	3DG	Spring	0.5 × 4 × 5	0.9	0.5
3DG + FN	3DG	FN	4	0.9	0.5
CP + S	CP	Spring	0.5 × 4 × 5	0.78	0.28
CP + FN	CP	FN	4	0.78	0.28

**Table 2 micromachines-10-00658-t002:** Comparison of B-μDMFC and other two μDMFC performance parameters.

Type	Area of μDMFC	Maximum Current Density	Maximum Power Density	Power Density per Unit Volume
μDMFC packaged by bolts [16]	4.84 cm^2^	30 mA cm^−2^	2.35 mW cm^−2^	2.94 mW cm^−3^
DMFC packaged by bolts [17]	72.25 cm^2^	40 mA cm^−2^	19.7 mW cm^−2^	5.57 mW cm^−3^
B-μDMFC	2.96 cm^2^	34 mA cm^−2^	4.78 mW cm^−2^	11.85 mW cm^−3^

**Table 3 micromachines-10-00658-t003:** Fitted parameters of the Nyquist plots.

Type of μDMFC	*R* _m_	*R* _ct_	*R* _co_	*R* _mt_	Maximum Power Density
3DG + S	0.45 Ω	1.82 Ω	3.52 Ω	3.6 Ω	4.78 mW cm^−2^
CP + S	0.54 Ω	3.80 Ω	0.61 Ω	6.2 Ω	4.20 mW cm^−2^
CP + FN	1.15 Ω	0.67 Ω	0.27 Ω	2.01 Ω	2.53 mW cm^−2^
3DG + FN	0.86 Ω	0.15 Ω	0.38 Ω	11.6 Ω	3.62 mW cm^−2^

**Table 4 micromachines-10-00658-t004:** Main geometric parameters and working parameters of the B-μDMFC model.

Parameters	Units	Values
Operating temperature T	K	300
Thickness of FN	mm	2.4
Thickness of 3DG	mm	0.16
Porosity of FN	-	0.59
Porosity of 3DG	-	0.7
Diffusion coefficient of methanol in water	m^2^ s^−1^	10^−5.4163−999.778/T^
Methanol penetration rate	mol cm^−2^ s^−1^	2 × 10^−7^
Discharge current *I*	mA cm^−2^	34.12
Methanol consumption flux	mol cm^−2^ s^−1^	2.6045 × 10^−7^
Methanol inlet concentration	mol L^−1^	1
*F*	C mol^−1^	96,485

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
