# Peer review of "A Novel Button-Type Micro Direct Methanol Fuel Cell with Graphene Diffusion Layer"

_micromachines, 2019, doi:10.3390/mi10100658_

Round 1

Reviewer 1 Report

The article titled "A Novel Button-type Micro Direct Methanol Fuel Cell with Graphene Diffusion Layer" deals with topic of making a new button-type micro direct methanol fuel cell (B-μDMFC). In this article, the introduction section and experimental part are well addressed, but some results are unclear. Therefore, I recommend this article for submission after Major revision, some remarks are listed below.

The authors missed all the material information including series number of the product and company. Please, provide detailed information. There are not detailed SEM images of 3DG, I recommend the authors to provide the images in the top view and cross-sectional view before and after assembly. When I saw Figure 6b, which is the SEM image of 3DG, there look ultra small pores for liquid and gas penetrating into the MEAs. What is (approximate) pore size of the 3DG after assembly? Figure 6a is not a SEM image of 3DG, rather it looks like metal foam. What is the meaning of ‘Before packaging’? Then, how did the authors measure the cell performance before packaging? There are some typos including “three dimension graphene” on line 15, “have” on line 75 and 76, “three structures cells” on line 85, “the” on line 138, etc.

Reviewer 2 Report

The authors designed button-type DMFCs with different components, and compared with each other and analyzed their operating characteristics. The button-type DMFC is considered to be highly feasible as a micro DMFC, but no examples have been studied. A structure incorporating a spring for reducing the contact resistance between the current collecting layer and the MEA is proposed, and the obtained volumetric power density exceeds the conventional one, which is considered to include an engineering value.

However, the structure of each component is not clearly shown, then the difference in the cell structure is difficult to understand. Also, there is no description of what properties 3DG has. It is also inappropriate that the details of the experimental conditions of the power generation experiment are not shown in Experimental section. The following points should be corrected.

1) Clarify the physical properties of CP, 3DG, S, and FN, that is, shape, thickness, porosity or opening ratio, electrical conductivity, etc., for four different cell structures. They are intended to show the difference in cell structure. Looking at Fig. 7 (b), it seems that FN is not a flat plate. It would be an idea to summarize these physical properties in a table.

2) Describe the details of the power generation experiment, that is, the methanol concentration and temperature used in Experimental.

3) Explain clearly what the packaging operation is and what the difference in the cell structure between before packaging and after packaging.

4) The scale should be shown in the SEM picture of Fig. 6.

5) Figs. 4 b and 4d should include simulation results calculated using the parameter values in Table 2.

6) The difference of the impedance before and after packaging shown in Fig. 4 and the difference of Rm between S and FN in Table 2 were affected by the difference in pressure applied between the MEA and the current collectors. I think that the contact resistance between them must be related to these differences. It should be argued that Rm includes contact resistance, and that the difference in contact resistance due to compression may be the cause of the difference in Rm between s and FN.

7) Something strange in English can be found, so correction by a native English speaker is required.

Round 2

Reviewer 1 Report

The authors have addressed all the comments I raised, so I suggest this article is accepted in present form.

Reviewer 2 Report

Based on the comments made in the previous review, appropriate revisions have been made, so this paper is judged to be acceptable.